# LUCID: Attention with Preconditioned Representations

## Abstract

Softmax-based dot-product attention is a cornerstone of Transformer architectures, enabling remarkable capabilities such as in-context learning. However, as context lengths increase, a fundamental limitation of the softmax function emerges: it tends to diffuse probability mass, assigning non-trivial weights to irrelevant tokens. This dilutes focus and degrades precision, especially in long-sequence scenarios. We introduce *LUCID Attention*, an architectural modification that applies a preconditioner to the attention probabilities. This preconditioner, derived from exponentiated key-key similarities, minimizes overlap between the keys in a Reproducing Kernel Hilbert Space, thus allowing the query to focus on important keys among large number of keys accurately. If a query $\mathbf{q}$ is highly similar to a key $\mathbf{k}$, LUCID outputs the corresponding value vector $\mathbf{v}$ with minimal blending from other tokens. This mechanism enables significantly sharper and more precise attention distributions. LUCID is designed as a drop-in replacement for existing attention mechanisms, retaining the same asymptotic complexity. We validate our approach by training $\sim 1$ billion parameter language models, pre-trained on a 2K sequence length and then fine-tuned up to a 65K sequence length. Our results demonstrate improved next-token prediction loss and significant gains on long-context retrieval tasks. LUCID shows an average improvement of $\sim 20\%$ in single and multi-needle in a haystack benchmarks compared to standard attention.

## 1 Introduction

Transformers, underpinned by the softmax dot-product attention mechanism, have become the cornerstone of modern machine learning, particularly in the domain of large language models (LLMs) (Vaswani et al., 2017). This mechanism, operating on queries (Q), keys (K), and values (V), allows models to dynamically weigh the importance of different parts of the input sequence. The exponential function within the softmax operation is crucial, enabling a sharp focus on relevant tokens, which forms the basis for remarkable capabilities such as in-context learning (Brown et al., 2020) and retrieval.

However, as the demand grows for LLMs to process increasingly longer sequences for tasks involving complex reasoning or extended documents, limitations of the standard attention mechanism become more apparent. Although softmax encourages focus, it often assigns non-negligible weight to irrelevant tokens, leading to diluted attention distributions. This phenomenon, often referred to as "attentional noise," can reduce precision and degrade performance on long-context tasks (Ye et al., 2025). Such issues are empirically observed as attention sinks (Gu et al., 2024), suggesting that standard attention lacks sufficient sparsity to scale gracefully with sequence length. Additionally, softmax's sensitivity to temperature can affect learability of the model, high temperature can lead to representation collapse (Masarczyk et al., 2025) and low temperature can lead to near zero Jacobian of the softmax operation.

Differential Transformer (Ye et al., 2025) induces sparsity by subtracting two softmax operations. Linear attention variants have shown that improvements are possible by introducing correction terms based on key structure. Works like DeltaNet (Yang et al., 2024b) and Gated DeltaNet (Yang et al., 2024a) demonstrate this in linearized attention settings by refining how attention aggregates values based on key similarity.

Inspired by these developments, we ask whether such correction strategies can benefit *standard softmax attention* by targeting the root cause of the attention noise issue - correlated keys. Viewing softmax attention through the lens of kernel methods (Katharopoulos et al., 2020), where attention probability is proportional to an inner product in a Reproducing Kernel Hilbert Space (RKHS) $\exp(\langle \mathbf{q}, \mathbf{k} \rangle) = \langle \phi(\mathbf{q}), \phi(\mathbf{k}) \rangle$. The crucial insight at the core of our paper is that we decorrelate the keys in RKHS feature space by preconditioning $\phi(\mathbf{k})$, in contrast to preconditioning the keys themselves as done in Gated DeltaNet and DeltaNet. This allows the queries in the feature space $\phi(\mathbf{q})$ to focus on selected keys among large number of keys with minimal noise, improving the retrieval performance.

**Contributions**:

1. We propose a novel preconditioning strategy that decorrelates keys in a Reproducing Kernel Hilbert Space (RKHS), which sharpens attention distributions and reduces noise from irrelevant tokens in long-context scenarios.

2. We demonstrate that our approach improves the conditioning of the softmax function, addressing critical learnability issues with softmax - vanishing gradient.

3. Our method achieves superior performance on needle in a haystack benchmarks, outperforming strong baselines such as the Differential Transformer (Ye et al., 2025), DeltaNet (Yang et al., 2024b) and Path Attention (Yang et al., 2025).

## 2 RELATED WORK

**Standard Attention and Long Context Challenges.** The scaled dot-product attention mechanism, introduced by Vaswani et al. (2017), is the cornerstone of the Transformer architecture. Its ability to model long-range dependencies and its parallelizability have driven the success of modern LLMs. The core computation involves Queries - $Q$, Keys - $K$, and Values - $V$, where attention probabilites are derived from $QK^T$ passed through a softmax function.

However, standard attention suffers from $\mathcal{O}(N^2 d)$ computational complexity and performance degradation on long sequences ($N$ large). This degradation manifests as "attention noise", where the softmax forces attention weights onto irrelevant tokens, obscuring the signal from relevant ones (Ye et al., 2025). This is particularly problematic for tasks requiring precise retrieval or reasoning over extended contexts. Another related phenomenon is the emergence of "attention sinks," often the first token (''), which attract disproportionately high attention regardless of semantic relevance (Gu et al., 2024). While potentially serving functional roles, sinks can also arise from softmax normalization constraints or training dynamics and contribute to noise.

**Noise Reduction and Sparsity.** Several approaches aim to mitigate attention noise and improve focus with quadratic computational complexity as standard attention. The Differential Transformer (Diff Transformer) (Ye et al., 2025) attempts to cancels noise by computing the difference between two attention maps ($A_1 - \lambda A_2$), promoting sparsity. There are methods which induce sparsity through fixed patterns, low-rank approximations (Han et al., 2023), or dynamic key selection via Approximate Nearest Neighbor Search (ANNS) like RetrievalAttention (Liu et al., 2024).

In contrast, LUCID attention tries to solve the root cause of the attention noise issue - correlated keys. In LUCID, a preconditioner developed from key-key similarities is used, which decorrelates the keys and removes attentional noise. This preconditioning sharpens the attention distribution and enable precise retrieval.

**Linear Complexity Alternatives.** Another line of research seeks to overcome the $\mathcal{O}(N^2)$ bottleneck by developing linear complexity ($\mathcal{O}(N)$) attention mechanisms. These often replace softmax with simpler kernels and reformulate the computation as a recurrent process, eliminating the need for a large KV cache (Katharopoulos et al., 2020; Yang et al., 2024b). While efficient, early linear attention variants often underperformed softmax attention, particularly in recall. DeltaNet (Yang et al., 2024b) improves recall within this linear framework by replacing the simple additive memory update with one inspired by the delta rule, allowing for more targeted memory modification. Gated-DeltaNet (Yang et al., 2024a) further enhances this by combining the delta rule with a data-dependent gating mechanism (similar to Mamba2 (Dao & Gu, 2024)) for adaptive memory erasure and targeted

updates. LUCID Attention is distinct from these approaches as it retains the $\mathcal{O}(N^2 d)$ complexity, focusing on improving the quality of standard attention mechanism rather than achieving efficiency. PaTH Attention Yang et al., 2025 introduces a data-dependent positional encoding mechanism via preconditioners, developed as a generatlization of the DeltaNet.

**Theoretical Perspectives.** Attention can be viewed through the lens of kernel methods (Katharopoulos et al., 2020). The softmax function $\exp(\langle \mathbf{q}, \mathbf{k} \rangle)$ corresponds to a positive-definite kernel that admits a (possibly infinite-dimensional) feature map $\phi$ into a Reproducing Kernel Hilbert Space (RKHS), such that $\exp(\langle \mathbf{q}, \mathbf{k} \rangle) = \langle \phi(\mathbf{q}), \phi(\mathbf{k}) \rangle$. LUCID Attention leverages this perspective by preconditioning the softmax output using a matrix derived from the masked exponentiated key-key inner products. This operation approximately inverts the attention smoothing effect caused by the kernel expansion and enables exact value retrieval when queries equal keys.

Table 1: Comparing mathematical formula of different attention mechanisms. Here, $Q, K, V \in \mathbb{R}^{N \times d}$ are queries, keys, and values, $\lambda \in \mathbb{R}, \boldsymbol{\beta} \in \mathbb{R}^N, W \in \mathbb{R}^{N \times d}$ are learnable parameters. $M$ and $\hat{M}$ are multiplicative and additive causal masks, respectively. $T = \mathrm{diag}(\boldsymbol{\beta})^{-1} + \mathrm{stril}(WW^\top)$. Let diag, tril, and stril be diagonal, lower-triangular and strictly-lower-triangular retention operators, respectively.

| Model | Attention Formula |
|---|---|
| **Standard** | $\mathrm{softmax}\left(QK^\top + \hat{M}\right) V$ |
| **Diff Transformer** | $\left(\mathrm{softmax}\left(Q_1 K_1^\top + \hat{M}\right) - \lambda \, \mathrm{softmax}\left(Q_2 K_2^\top + \hat{M}\right)\right) V$ |
| **DeltaNet** | $\left(M \circ QK^\top\right)\left(I + \mathrm{diag}(\boldsymbol{\beta})\mathrm{stril}(KK^\top)\right)^{-1} \mathrm{diag}(\boldsymbol{\beta})V$ |
| **PaTH** | $\mathrm{softmax}\left(\mathrm{tril}(QK^\top) - \mathrm{tril}(QW^\top)T^{-1}\mathrm{stril}(WK^\top)\right) V$ |
| **LUCID (Ours)** | $\mathrm{softmax}\left(QK^\top + \hat{M}\right)\left(M \circ \exp\left(KK^\top\right)\right)^{-1} V$ |

# 3 LUCID ATTENTION

We begin by formally introducing the standard softmax attention mechanism, followed by its linear variant. We then identify a fundamental retrieval problem that exposes a limitation of softmax attention. To resolve this, we propose LUCID Attention: a method that preconditions the attention logits for improved retrieval. Finally, we describe in detail, the connections between LUCID Attention and prior work in the literature.

## 3.1 LIMITATIONS OF STANDARD SOFTMAX ATTENTION

We first start by writing the equation of causal softmax attention. Let $Q, K, V \in \mathbb{R}^{N \times d}$ denote the query, key, and value matrices, where $N$ is the sequence length and $d$ is the head dimension. Each row $\mathbf{q}_i$, $\mathbf{k}_j$, and $\mathbf{v}_j$ in the matrices $Q, K, V$ corresponds to the $i$-th or $j$-th query, key, or value vector, respectively.

Let the causal mask $M \in \{0, 1\}^{N \times N}$, $M_{ij} = 1$ if $i \geq j$ and $M_{ij} = 0$ otherwise. Additionally, we define $\hat{M} \in \{0, -\infty\}^{N \times N}$, where $\hat{M}_{ij} = 0$ if $i \geq j$ and $\hat{M}_{ij} = -\infty$ otherwise. Alternatively, the binary mask $M$ can be written as $M = \exp(\hat{M})$.

The standard softmax attention with a causal mask is given by:

$$S = \mathrm{softmax}\left(QK^\top + \hat{M}\right) V,$$

$$\mathrm{softmax}(A)_{ij} = \frac{\exp(A_{ij})}{\sum_{j=1}^{s} \exp(A_{ij})}$$

where $S \in \mathbb{R}^{N \times d}$ is the output matrix.

Focusing on a single row $\mathbf{s}_i \in \mathbb{R}^d$, we can expand the softmax expression as:

$$\mathbf{s}_i = \sum_{j=1}^{i} \frac{\exp\left(\langle \mathbf{q}_i, \mathbf{k}_j \rangle\right)}{\sum_{j'=1}^{i} \exp\left(\langle \mathbf{q}_i, \mathbf{k}_{j'} \rangle\right)} \mathbf{v}_j,$$

where the summation is restricted to positions $j \leq i$ due to the causal mask.

Following (Katharopoulos et al., 2020), the exponential inner product can be expressed using a kernel function:

$$\exp\left(\langle \mathbf{q}_i, \mathbf{k}_j \rangle\right) = \langle \phi(\mathbf{q}_i), \phi(\mathbf{k}_j) \rangle,$$

where $\phi : \mathbb{R}^d \to \mathcal{H}$ is a feature map mapping each row vector onto a Reproducing Kernel Hilbert Space (RKHS).

Extending this to matrices, we apply $\phi$ row-wise to obtain $\phi(Q), \phi(K) \in \mathbb{R}^{N \times r}$, leading to the generalized kernel attention:

$$S = D^{-1} \left( M \circ \left( \phi(Q)\phi(K)^\top \right) \right) V, \quad D_{ii} = \sum_{j=1}^{i} \langle \phi(\mathbf{q}_i), \phi(\mathbf{k}_j) \rangle,$$

Here, $\circ$ denotes element-wise multiplication. In the above equation, by setting $\phi(x) = \text{elu}(x) + 1$, i.e., using the identity feature map, we recover linear attention as proposed in (Katharopoulos et al., 2020).

**Can standard attention represent sharp distributions?** This can be achieved in softmax in the zero-temperature limit, consider a vector $\mathbf{z} \in \mathbb{R}^N$, then:

$$\text{softmax}(\mathbf{z}/\tau) \xrightarrow{\tau \to 0} \mathbf{e}_{\arg \max \mathbf{z}}, \tag{1}$$

where $\mathbf{e}_i$ is the canonical basis vector or one hot vector with $i$-th element as 1 and rest set to 0. However, this introduces a major problem: the Jacobian of softmax vanishes at saturation. In particular, the softmax Jacobian is:

$$\mathbf{a} = \text{softmax}(\tilde{\mathbf{a}}),$$

$$\frac{\partial \mathbf{a}}{\partial \tilde{\mathbf{a}}} = J = \text{diag}(\mathbf{a}) - \mathbf{a}\mathbf{a}^\top.$$

If $\mathbf{a} = \mathbf{e}_i$, then

$$J = \text{diag}(\mathbf{e}_i) - \mathbf{e}_i \mathbf{e}_i^\top = 0.$$

This implies that zero gradients are propagated, thus making learning ineffective.

These limitation motivate the need for attention mechanisms that can represent sharp attention distributions bypassing the learnability issue with softmax attention.

## 3.2 LUCID ATTENTION OFFERS PRECISE RETRIEVAL OF VALUES

We now ask: what matrix $P \in \mathbb{R}^{N \times N}$ can be right-multiplied to softmax logits so that, when queries equal keys ($Q = K$), the resulting attention distribution is close to identity? That is, we wish to find $P$ such that the problem

$$(M \circ \exp(QK^\top))P = I_N$$

is solved when $Q := K$, where $M \in \{0,1\}^{N \times N}$ is the binary causal mask and $\exp(\cdot)$ is the element-wise exponential function. Solving the linear system yields:

$$P = \left( M \circ \exp(KK^\top) \right)^{-1},$$

The resulting attention mechanism, which we refer to as LUCID Attention, guarantees that when $\mathbf{q}_i = \mathbf{k}_i$, the retrieved value is $\mathbf{v}_i$, up to a scaling factor. Importantly, the causal mask $M$ ensures that $M \circ \exp(KK^\top)$ is lower triangular with strictly non-zero diagonal entries, guaranteeing invertibility.

This triangularity, induced by masking future positions, ensures the matrix has full rank and non-zero eigenvalues.

To demonstrate this explicitly, consider a query $\mathbf{q}_t = \mathbf{k}_t$ corresponding to a specific key $\mathbf{k}_t$. Then the attention output is:

$$\text{LUCID}(\mathbf{q}_t, K, V) = \frac{\mathbf{m}_t \circ \exp(\mathbf{q}_t^\top K)}{\sum_{j=1}^{i} \exp(\mathbf{q}_t^\top \mathbf{k}_j)} \left( M \circ \exp(KK^\top) \right)^{-1} V,$$

where $\mathbf{m}_t$ is the $t$-th row of $M$. Substituting $\mathbf{q}_t = \mathbf{k}_t$, the numerator becomes the $t$-th row of $M \circ \exp(KK^\top)$, and the entire expression recovers $\mathbf{v}_t$, upto a scale $1/\sum_{j=1}^{t} \exp(\mathbf{q}_t^\top \mathbf{k}_j)$.

While this formulation retrieves values precisely, we must also ensure that the scale of activations remains consistent with standard softmax attention, as emphasized in the literature on stable training dynamics (e.g., (Yang et al., 2022)).

So far, we have omitted the logit scaling factor $1/\sqrt{d}$ for clarity. In practice, standard softmax attention applies:

$$\text{softmax}\left( \frac{QK^\top}{\sqrt{d}} + \hat{M} \right),$$

where $d$ is the head dimension. To maintain numerical stability and proper variance scaling, we introduce RMS normalization for the key vectors inside the exponential:

$$\mathbf{k}_{i,\text{RN}} \leftarrow \sqrt{d} \cdot \mathbf{k}_i / \|\mathbf{k}_i\|_2.$$

This normalization ensures that the inner products $\mathbf{k}_{i,\text{RN}}^\top \mathbf{k}_{j,\text{RN}}$ have the same distributional scale across tokens and the exponential matrix is unit-diagonal and diagonal-heavy for a better condition number. This approach is closely related to QK-Norm (Dehghani et al., 2023), a technique in which LayerNorm is applied directly to $Q$ and $K$ vectors before computing attention weights, except, we only normalize the keys passed to the preconditioner.

Putting all components together, the LUCID Attention can be expressed as:

$$\text{LUCID}(Q, K, V) = \left( \text{softmax}\left( \frac{QK^\top}{\sqrt{d}} + \hat{M} \right) \right) \left( M \circ \exp\left( \frac{K_{\text{RN}} K_{\text{RN}}^\top}{\sqrt{d}} - \sqrt{d} \right) \right)^{-1} V. \quad (2)$$

This final formulation recovers softmax-like attention weights but guarantees precise retrieval when $Q = K$, while also remaining differentiable during training. The normalization inside the exponential won't affect the retrieval ability as the matrix $K_{\text{RN}} K_{\text{RN}}^\top$ is symmetric so the left-multiplication with the softmax matrix with $\mathbf{q}_t = \mathbf{k}_t$ will still return one-hot vector. In subsequent sections, we evaluate its empirical behavior across different sequence lengths and head dimensions.

LUCID Attention doesn't suffer from vanishing gradient problem as temperature controlled softmax attention in (1). The Jacobian with respect to $\mathbf{q}$ is:

$$\mathbf{o} = \text{softmax}\left( \frac{\mathbf{q}K^\top}{\sqrt{d}} \right) \left( M \circ \exp\left( \frac{K_{\text{RN}} K_{\text{RN}}^\top}{\sqrt{d}} - \sqrt{d} \right) \right)^{-1}$$

$$= \mathbf{a} \left( M \circ \exp\left( \frac{K_{\text{RN}} K_{\text{RN}}^\top}{\sqrt{d}} - \sqrt{d} \right) \right)^{-1},$$

$$\frac{\partial \mathbf{o}}{\partial \mathbf{q}} = \frac{K^\top}{\sqrt{d}} \left( \text{diag}(\mathbf{a}) - \mathbf{a}\mathbf{a}^\top \right) \left( M \circ \exp\left( \frac{K_{\text{RN}} K_{\text{RN}}^\top}{\sqrt{d}} - \sqrt{d} \right) \right)^{-1}.$$

Because the softmax output $\mathbf{a}$ is not one-hot, the gradient with respect to $\mathbf{q}$ stays non-zero and learning does not stagnate under an assumption that at-least one column of $\text{diag}(\mathbf{a}) - \mathbf{a}\mathbf{a}^\top$ is not in the null-space of $K^\top$ (proof in appendix). A similar analysis for keys can be done, however, we are not showing it here.

In summary, we show that LUCID Attention has precise retrieval similar to zero-temperature softmax and non-vanishing gradients passing through queries.

## 3.3 LUCID'S EFFICIENCY

Implementing LUCID Attention requires solving a triangular linear system involving the masked kernel matrix. Specifically, we solve for $Y \in \mathbb{R}^{N \times d}$ in:

$$\left( M \circ \exp\left( \frac{K_{\mathrm{RN}} K_{\mathrm{RN}}^\top}{\sqrt{d}} - \sqrt{d} \right) \right) Y = V,$$

which can be done efficiently via forward substitution, since the matrix is lower triangular due to the causal mask $M$. Once $Y$ is obtained, the final output is computed by multiplying the softmax attention weights:

$$\mathrm{LUCID}(Q, K, V) = \left( \mathrm{softmax}\left( \frac{QK^\top}{\sqrt{d}} + \hat{M} \right) \right) Y.$$

The overall computational complexity remains $\mathcal{O}(N^2 d)$, similar to standard softmax attention.

## 3.4 RELATION TO KERNEL AND DELTANET-BASED ATTENTION MECHANISMS

Several recent works have proposed mechanisms to improve attention expressiveness and retrieval quality by modifying the structure of attention weights or the space in which attention is computed. In this section, we relate LUCID Attention to such approaches.

Recall from Section 3.1 that the exponential kernel used in softmax attention can be written as:

$$\exp(\langle \mathbf{q}_i, \mathbf{k}_j \rangle) = \langle \phi(\mathbf{q}_i), \phi(\mathbf{k}_j) \rangle,$$

for feature map $\phi : \mathbb{R}^d \to \mathcal{H}$. This correspondence is guaranteed by the Moore–Aronszajn theorem (Aronszajn, 1950) for positive-definite kernels. Note that, the exponential kernel induces $\phi(\cdot)$, which is an infinite-dimensional feature map. Expanding this idea to matrix form gives the general kernel attention:

$$O = D^{-1} \left( M \circ \left( \phi(Q) \phi(K)^\top \right) \right) V,$$

where $D$ handles the row-wise normalization defined in Section 3.1.

Setting $\phi(x) = x$, i.e., the identity map, recovers linear attention, where the kernel is the standard dot product and attention weights are not exponentiated. A closely related line of work is DeltaNet (Yang et al., 2024b), which proposes a preconditioned variant of attention based on parameterized lower-triangular corrections. The DeltaNet attention mechanism is given by:

$$\left( M \circ QK^\top \right) \left( I_N + \mathrm{stril}\left( \mathrm{diag}(\boldsymbol{\beta}) KK^\top \right) \right)^{-1} \mathrm{diag}(\boldsymbol{\beta}) V,$$

where $\boldsymbol{\beta} = (\beta_1, \ldots, \beta_N)$ are learned per-token scaling parameters, and the stril operator zeroes the diagonal and upper triangle. If we set $\boldsymbol{\beta} = \mathbf{1}$ and introduce a kernel feature map $\phi$, DeltaNet reduces to a form similar to LUCID Attention. In our experiments, we did not find performance gains from learning $\boldsymbol{\beta}$.

The recursive equation for DeltaNet in the feature space with $\boldsymbol{\beta} = \mathbf{1}$ is

$$\mathbf{S}_t = \mathbf{S}_{t-1}(\mathbf{I} - \phi(\mathbf{k}_t)\phi(\mathbf{k}_t)^\top) + \mathbf{v}_t \phi(\mathbf{k}_t)^\top,$$
$$\mathbf{o}_t = \mathbf{S}_t \mathbf{q}_t$$

where $\mathbf{S}_0 = \mathbf{0}$. This structure facilitates a memory rewriting mechanism where past values are updated using the current key, which is designed to retain more unique information for retrieval tasks. In DeltaNet, since the feature map is the identity, this memory rewriting is limited to a finite-dimensional representation.

In contrast, LUCID Attention operates in an infinite-dimensional kernel space, which provides greater expressivity than the token space used by DeltaNet. This allows for a more optimal memory rewriting process and improves the retention of unique information. The preconditioner $(M \circ \exp(KK^\top))^{-1}$ can be viewed as a causal de-correlating operator that removes redundant information, ensuring that more new information is learned at each step.

## 4 EXPERIMENTAL SETUP

We compare LUCID Attention against different baselines: Standard Attention, Diff Transformer, DeltaNet, and PaTH Attention. We train, fine-tune, and evaluate all models on a subset of Dolma dataset ($\sim$6.5B tokens) provided by Allen AI Soldaini et al., 2024 and use it for Needle-In-A-Haystack (NIAH) evaluations and downstream tasks.

### 4.1 EXPERIMENTAL SETUP

For NIAH evaluations and downstream tasks, we train decoder-only transformers with 22 layers, model dimension 2048, MLP dimension 5632, and vocabulary size 32,000. Each model uses 32 attention heads and 4 key-value heads with head dimension of 68. PaTH Rotary positional embeddings - RoPE (Su et al., 2024) are used in all attention variants except for DeltaNet and PaTH. RoPE frequencey is set to 10,000 during pretraining and was changed to 64,000 during fine-tuning. We use the AdamW optimizer (Diederik, 2014) with learning rate $5 \times 10^{-4}$, $\beta_1 = 0.9$, and $\beta_2$=0.99 with weight decay of $0.01$. We conduct pretraining of all models on sequence length 2048 for 11k optimization steps with each step having effective batch size of 256 sequences. Then we fine-tune on sequence length 65536 for 500 optimization steps with each step having effective batch size of 16. In both, pre-training and fine-tuning, LUCID Attention's wall-clock time was comparable to that of Standard Attention. We use Huggingface Transformers library Wolf et al., 2020 to train our models.

We also conduct some log-perplexity based evaluations at larger scale by training a 16 layer model with model dimension 2048, feed-forward dimension 4096, vocabulary size 256,000, and $\beta_2 = 0.95$ for Adam. We trained the models on sequence lengths 4096 and 32,768. For the 4096-token setup, we train with a batch size of 1M tokens per iteration for 80k steps (80B tokens). For the 32K-token setting, we use 2M tokens per iteration for 25k steps (50B tokens).

### 4.2 NEEDLE-IN-A-HAYSTACK EVALUATIONS

We conduct experiments on SNIAH and MNIAH tasks before and after fine-tuning on varying sequence lengths and number of needles. For these tasks, we modified the RULER tasks Hsieh et al., 2024 in the eval-harness framework Gao et al., 2024. Note that RULER has multiple SNIAH and MNIAH tasks. Here, we are reporting average accuracies.

Table 2: Performance comparison of different models against LUCID Attention for MNIAH task at 2048 sequence length.

| Model | Number of Needles | | | | |
|---|---|---|---|---|---|
| | 2 | 4 | 6 | 8 | 10 |
| **Standard** | 74.2 | 51.0 | 38.8 | 30.6 | 24.8 |
| **Diff Transformer** | 72.4 | 35.4 | 26.0 | 20.2 | 14.8 |
| **DeltaNet** | 37.0 | 16.8 | 11.0 | 7.4 | 3.2 |
| **Path** | 61.4 | 44.2 | 37.2 | 30.4 | **26.0** |
| **LUCID** | **76.6** | **55.8** | **43.6** | **33.8** | 25.2 |

Table 3: Performance comparison of Standard Attention and LUCID Attention models for SNIAH and MNIAH tasks across different sequence lengths after fine-tuning.

| Task | Model | Sequence Length | | | | | | |
|---|---|---|---|---|---|---|---|---|
| | | 2048 | 4096 | 6144 | 8192 | 16384 | 32768 | 65536 |
| **SNIAH** | **Standard** | 55.9 | 47.5 | 32.5 | 19.3 | 2.0 | 0.0 | 2.0 |
| | **LUCID** | 62.2 | 68.6 | 68.2 | 50.2 | 15.0 | 1.0 | 2.0 |
| **MNIAH** | **Standard** | 33.6 | 37.4 | 26.0 | 23.2 | 4.0 | 4.0 | 2.0 |
| | **LUCID** | 51.6 | 51.4 | 40.4 | 33.2 | 14.0 | 12.0 | 2.0 |

Table 4: Performance comparison of Standard Attention and LUCID Attention models for MNIAH task for varying number of needles and sequence lengths.

| Needles | Model | Sequence Length | | | |
|---|---|---|---|---|---|
| | | 2048 | 4096 | 6144 | 8192 |
| 2 | **Standard** | 57.0 | 59.8 | 37.4 | 21.8 |
| | **LUCID** | 69.0 | 68.6 | 57.0 | 48.0 |
| 4 | **Standard** | 33.6 | 37.4 | 26.0 | 23.2 |
| | **LUCID** | 51.6 | 51.4 | 40.4 | 33.0 |
| 6 | **Standard** | 26.8 | 20.4 | 20.4 | 22.4 |
| | **LUCID** | 40.2 | 33.4 | 33.0 | 31.6 |
| 8 | **Standard** | 17.6 | 19.0 | 17.4 | 17.0 |
| | **LUCID** | 27.6 | 27.6 | 27.6 | 24.2 |
| 10 | **Standard** | 15.8 | 15.2 | 12.8 | 11.4 |
| | **LUCID** | 22.8 | 23.2 | 23.2 | 23.4 |

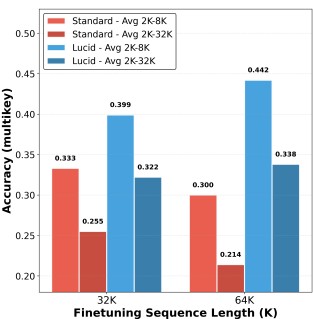

Figure 1: **Multi-needle retrieval accuracy improves with longer finetuning for LUCID.** Models finetuned at 32K and 64K sequence lengths are evaluated on multi-needle tasks with contexts averaged over 2K-8K and 2K-32K. The performance gap between LUCID and Standard Attention increases from +19.8% (32K finetuning) to +47.3% (64K finetuning), indicating that LUCID's benefits scale positively with finetuning sequence length while Standard Attention degrades at ultra-long contexts.

Throughout the NIAH evaluations, LUCID Attention shows better retrieval capability compared to Standard Attention both for varying sequence lengths and varying number of needles. we also vary the finetuning sequence length and observe that LUCID scales better with finetuning sequence length compared to standard attention.

## 4.3 INCREASING PRETRAINING CONTEXT LENGTH

We first evaluate how LUCID performs as sequence length increases while holding attention head size constant. At 4096 context length, LUCID improves training loss by 0.012 over the Transformer baseline after 80B tokens. At 32,768 context length, the gain increases to 0.048 under the same architecture and optimization settings. These results support our hypothesis: for a fixed representation size, the need to precondition grows with sequence length. As standard softmax attention becomes increasingly diffuse over longer contexts, LUCID corrects this by concentrating probability mass on relevant tokens, thereby enhancing retrieval capability.

## 4.4 ABLATIONS

**Reduced Head Size.** We next examine the effect of reducing the head dimension while keeping the total model dimension constant. Specifically, we fix the sequence length to 4096 from the experiment in Section 4.3 and reduce the per-head size from 64 to 16, increasing the number of heads proportionally from 32 to 128.

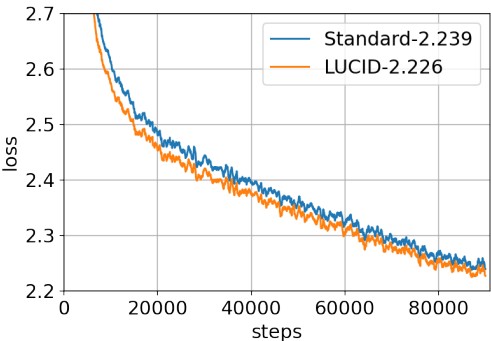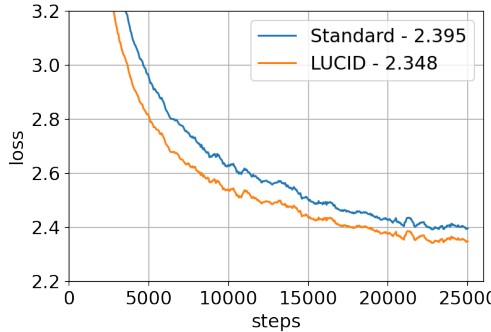

Figure 2: Training loss vs. steps for Standard Attention vs. LUCID Attention at different context lengths. LUCID achieves lower validation loss at both 4096 (left) and 32768 (right) sequence lengths, indicating improved retrieval in long-context settings.

Under this setup, LUCID attention achieves a 0.1 improvement in validation loss over the Transformer baseline for headsize 16. This confirms our intuition that LUCID is most effective when the attention representation is narrow relative to the sequence length. In this regime, softmax attention lacks the capacity to sharply resolve relevant keys, while LUCID compensates through structural preconditioning.

**RoPE.** Rotary Position Embeddings (RoPE) introduce relative position information by applying structured rotation matrices to the query and key vectors. This results in higher cosine similarity between nearby positions and a natural decay in similarity as the relative distance increases. We have found that the loss difference between LUCID (2.42) and standard attention (2.54) increases to 0.12 when the RoPE is removed in sequence length 32k setting (Figure 2). We conjecture that RoPE structures the keys and queries there by conditioning them for better retrieval. Absence of RoPE, alternatively called NoPE, is used in Llama 4 (Meta AI, 2024).

**LUCID Variants.** We test different LUCID variants on MNIAH task for different number of needles for 2048 sequence length obtained by experimenting with $\exp$ stabilization. Mainly we considered 1) key normalization: RMSNorm over the keys before preconditioning (2), 2) QK-Norm: RMSNorm over both queries and keys, 3) max normalization: dividing each row of $\mathrm{tril}(\exp(KK^\top))$ with the row maximum. For key normalization, we also consider learning $\beta$ as discussed in Section 3.4. The results are summarized in Table 5.

Table 5: Performance comparison of different variants of LUCID Attention for MNIAH task at 2048 sequence length.

| Variant | Number of Needles | | | | |
|---|---|---|---|---|---|
| | **2** | **4** | **6** | **8** | **10** |
| **key normalization ($\beta = 1$)** | **76.6** | **55.8** | **43.6** | **33.8** | 25.2 |
| **QK-Norm** | 75.4 | 51.4 | 37.2 | 32.0 | 20.4 |
| **max normalization** | 74.2 | 48.8 | 37.2 | 31.4 | 19.6 |
| **key normalization (learnable $\beta$)** | 73.8 | 48.0 | 36.8 | 31.8 | **26.8** |

## 5 CONCLUSION

LUCID Attention introduces an effective modification to standard softmax attention by preconditioning with key-key similarity matrix, significantly improving attention focus while preserving full $\mathcal{O}(N^2 d)$ complexity. Unlike prior methods that trade off expressivity for efficiency or require additional trainable components, LUCID enhances retrieval precision and sparsity without altering the core Transformer architecture. This makes it a drop-in improvement for long-context tasks, offering better focus and interpretability.

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

## A APPENDIX

### A.1 PROOF OF NON-VANISHING GRADIENTS

The Jacobian is:

$$\frac{\partial \mathbf{o}}{\partial \mathbf{q}} = \underbrace{\frac{K^\top}{\sqrt{d}} \left( \mathrm{diag}(\mathbf{a}) - \mathbf{a}\mathbf{a}^\top \right)}_{\text{Gradient of softmax w.r.t } \mathbf{q}} \left( M \circ \exp\left( \frac{K_{\mathrm{RN}}K_{\mathrm{RN}}^\top}{\sqrt{d}} - \sqrt{d} \right) \right)^{-1}.$$

As the preconditioner is invertible, its null-space is trivial. So, the gradient can only vanish if the gradient of softmax w.r.t $\mathbf{q}$ is zero. This can happen in three cases: 1) output is one-hot vector, 2) $K$ is trivially zero-matrix, and 3) all columns of $\mathrm{diag}(\mathbf{a}) - \mathbf{a}\mathbf{a}^\top$ lies in the null-space of $K^\top$. Because LUCID Attention doesn't use temperature setting, the softmax output is not one-hot and not considering the trivial case of $K$ being a zero-matrix, the only case remains is that all columns of $\mathrm{diag}(\mathbf{a}) - \mathbf{a}\mathbf{a}^\top$ lies in the null-space of $K^\top$. This concludes that LUCID Attention has non-vanishing gradients like scaled dot product attention while having precise retrieval of zero-temperature limit softmax.

### A.2 KV CACHING

To enable LUCID for auto-regressive behavior, we need to cache unnormalized keys $K_{\mathrm{past}} \in \mathbb{R}^{L_{\mathrm{past}} \times d}$ and the solution of the triangular solver $(M \circ \exp(K_{\mathrm{RN(past)}}K_{\mathrm{RN(past)}}^\top / \sqrt{d} - \sqrt{d}))Y_{\mathrm{past}} = V_{\mathrm{past}}$. When new keys $K_{\mathrm{new}} \in \mathbb{R}^{L_{\mathrm{new}} \times d}$ and values $V_{\mathrm{new}} \in \mathbb{R}^{L_{\mathrm{new}} \times d}$ are generated during decoding, new solutions of the triangular system $Y_{\mathrm{new}} \in \mathbb{R}^{L_{\mathrm{new}} \times d}$ can be easily computed by solving smaller triangular system

$$\left( M \circ \exp\left( \frac{K_{\mathrm{RN(new)}}K_{\mathrm{RN(new)}}^\top}{\sqrt{d}} - \sqrt{d} \right) \right) Y_{\mathrm{new}} = V_{\mathrm{new}} - \exp\left( \frac{K_{\mathrm{RN(new)}}K_{\mathrm{RN(past)}}^\top}{\sqrt{d}} - \sqrt{d} \right) Y_{\mathrm{past}}.$$

Once $Y_{\mathrm{new}}$ is obtained, the new outputs of LUCID Attention can be computed through standard softmax KV caching mechanism. Asymptotically, the auto-regressive complexity of LUCID Attention is same as Standard Attention.

### A.3 MORE EXPERIMENTS

We conduct downstream evaluation on checkpoints at the end of the training and measure accuracy.

Table 6: Performance comparison between LUCID attention and standard softmax attention across a range of downstream tasks.

|  | COPA | HellaSwag | MultiRC | PIQA | RACE-m | ReCoRD | StoryCloze | WiC | Winogrande | Overall |
|---|---|---|---|---|---|---|---|---|---|---|
| **LUCID** | 57.00 | 32.85 | 55.01 | 60.72 | 34.12 | 65.34 | 56.33 | 47.34 | 50.51 | **51.03** |
| **Standard** | 54.00 | 31.90 | 53.28 | 59.68 | 33.01 | 63.41 | 56.33 | 47.02 | 48.46 | **49.68** |

**Evaluation Suite.** We evaluate LUCID attention across a diverse suite of natural language understanding tasks, reflecting commonsense reasoning, reading comprehension, and sentence inference. COPA (Roemmele et al., 2011) tests causal reasoning. HellaSwag (Zellers et al., 2019) assesses commonsense story completion. MultiRC (Khashabi et al., 2018) involves multi-sentence question answering with multiple correct answers. PIQA (Bisk et al., 2020) evaluates physical commonsense. RACE-m (Lai et al., 2017) focuses on middle-school level reading comprehension. ReCoRD (Zhang et al., 2018) involves cloze-style passage understanding. StoryCloze (Mostafazadeh et al., 2016) tests narrative coherence. WiC (Pilehvar & Camacho-Collados, 2018) is a word sense disambiguation benchmark. Winogrande (Sakaguchi et al., 2020) evaluates coreference resolution under adversarial conditions. These tasks collectively stress various long-context abilities and highlight the precision improvements enabled by LUCID attention.

