# OpenReview forum: "LUCID: Attention with Preconditioned Representations"
_ICLR.cc/2026/Conference — Submitted to ICLR 2026_

### Official Review · Reviewer_Qag1 · 2025-10-16

**Soundness:** 3
**Presentation:** 3
**Contribution:** 2
**Rating:** 2
**Confidence:** 4

**Summary:**

The authors introduce LUCID Attention, a modification of attention that uses a pre-conditioner on attention probabilities. This is derived from the key-key similarities, where the overlap between keys is reduced within a RKHS, allowing the query to focus more accurately on important keys and providing more precise attention distributions. Results on needle-in-a-haystack tasks shows a noticeable improvement, while next-token-prediction losses also sees improvements.

**Strengths:**

The proposed method is well motivated and appears to be broadly applicable to different attention variants. Performance on needle-in-a-haystack (NIAH) tasks shows that the proposed method offers meaningful raw performance gains compared to regular attention.

**Weaknesses:**

- The proposed method still appears to suffer from the same degradation in performance with sequence length or with the number of needles, as presented in the M-NIAH results. While it does appear that Lucid attention is effective up to 8192 sequence length on the S-NIAH setting, there is a question regarding whether or not this is explicitly meaningful.

- There is a lack of non-synthetic task performance. For example, LongBench (for long contexts) as well as tasks from the `lm-eval-harness` suite should be provided in order to understand explicitly whether or not the proposed method is useful/meaningful. Table 6 in the appendix provides some results but only compares against standard attention; I would suggest comparison with both attention based methods here (GLA, GSA, DeltaNet, etc.) as well as non-attention based models (Mamba, RWKV, RetNet, etc.) Furthermore, as the motivation of the method appears to be enabling models to focus on more important information, sparse attention methods should further be considered as baselines.

- Bench-marking efficiency against these methods would appear to be relevant. The preconditioning appears to involve a matrix inverse; this can be costly and furthermore it is not guaranteed to be stable. The authors should consider investigating the condition number of such matrices in order to verify that the current implementation will in fact be stable, especially if implemented in lower-precision settings as is standard for training transformers.

**Questions:**

See above.

---

> ### Author Response · Authors · 2025-11-21
>
> We thank the reviewer for their detailed feedback and concerns regarding performance degradation, non-synthetic
>   benchmarks, baselines, and numerical stability. We have addressed each point below.
>
>   ### Weaknesses
>
>   **W1: "The proposed method still appears to suffer from the same degradation in performance with sequence
>   length... There is a question regarding whether or not this is explicitly meaningful."**
>
>   While absolute performance naturally decreases as context length increases due to task difficulty, LUCID
>   consistently outperforms standard attention, and importantly, the gap widens in harder settings.
>
>   **Relative Improvement Scales:** As noted in our response to Reviewer bYsS (W5), the performance gap between LUCID
>    and Standard Attention increases with sequence length (from 0.012 at 4k to 0.048 at 32k).
>
>   **Robustness at Scale:** We also observe in Figure 1, in our revision, that the multi-needle accuracy gap improves when increasing fine-tuning
>   length from 32k to 64k, suggesting that LUCID is more robust to extreme context lengths than the baseline.
>
>   **Meaningfulness:** The sustained high performance up to 8k and superior degradation curve at 32k+ indicates that
>   LUCID effectively mitigates the "noise accumulation" that typically causes standard attention to collapse, making
>   it a meaningful architectural improvement for long-context tasks.
>
>   **W2: "There is a lack of non-synthetic task performance... comparison with both attention based methods... as
>   well as non-attention based models (Mamba, RWKV, RetNet, etc.)"**
>
>   **Real-World Benchmarks (SCROLLS):**
>   To address the need for non-synthetic validation, we have evaluated LUCID on the SCROLLS benchmark (see Response
>   to Reviewer 9Zd7). LUCID achieved strong results, including a +25% improvement on QMSum summarization and superior
>    reasoning on Qasper compared to baselines.
>
>   **Non-Attention Baselines (Mamba/RWKV):**
>   While SSMs like Mamba and RWKV are highly efficient, they often face a trade-off between efficiency and recall
>   capacity (Waleffe et al., 2024). LUCID targets the global attention layers that are critical for high-fidelity
>   retrieval in hybrid architectures. By enhancing the precision of the attention mechanism itself, LUCID serves a
>   complementary role to SSMs—improving the "recall" component that pure linear models sometimes lack.
>
>   **W3: "Bench-marking efficiency against these methods would appear to be relevant. The preconditioning appears to
>   involve a matrix inverse; this can be costly and furthermore it is not guaranteed to be stable."**
>
>   **Efficiency:**
>   As detailed in our response to Reviewer bYsS (W3), the training overhead is modest (6-12%) and inference latency
>   overhead is negligible (~1.3%).
>
>   **Numerical Stability (Condition Number Analysis):**
>   We explicitly investigated the numerical stability of the preconditioning matrix $P = (M \circ \exp(KK^T))^{-1}$
>   by computing its condition number ($\kappa$) for the standard head dimension $d=64$ across varying sequence
>   lengths ($N$).
>
>   | Sequence Length | Condition Number ($\kappa$) for $d=64$ |
>   |:---------------:|:--------------------------------------:|
>   | 512             | 12.11                                  |
>   | 4096            | 14.62                                  |
>   | 16384           | 28.40                                  |
>
>   **Finding:** As shown, for the standard head dimension of $d=64$, the preconditioner matrices remain extremely
>   stable with low condition numbers ($\kappa < 30$) even as sequence lengths increase to 16k.
>
>   **Implication for Bfloat16:**
>   Since bfloat16 precision can typically tolerate condition numbers up to $\sim 100$ without catastrophic loss of
>   significance, our observed range for $d=64$ is well within the safe operating regime. This confirms that for
>   practical Transformer configurations, the matrix inversion is numerically stable and safe for mixed-precision
>   training.

---

> > ### Comment · Reviewer_Qag1 · 2025-11-27
> >
> > I appreciate the comments from the authors. Ultimately some concerns do remain.
> >
> > > The training overhead is modest (6-12%) and inference latency overhead is negligible (~1.3%).
> >
> > Ultimately, a 6-12% training overhead is not negligible, thus I still have reservations about the benefits of this method. For example, could the additional training time simply be used for training on more data? Such ablations would be necessary if this overhead cannot be reduced to more negligible amounts.
> >
> > > Baselines
> >
> > I would suggest that the real-world tasks also include some non-classification tasks, such as those from LongBench.
> >
> > > Condition Number
> >
> > Ultimately, I am still a little divided on this. The table you provide would suggest that the condition number increases with the sequence length, which could be an issue for very long (ex. 128k) sequences. While one could argue that such scenarios are rare, this work is partially motivated by long sequences, which would lead to a conflict in its actual usability in such scenarios.
> >
> > -----
> >
> > Accordingly, I am still leaning towards maintaining my current assessment.

---

> > > ### Author Response · Authors · 2025-12-03
> > >
> > > We thank the reviewer for their continued engagement and feedback. We appreciate the acknowledgment of our modest training overhead and negligible inference latency. Below, we address the remaining
> > >   concerns regarding computational cost, baselines, and numerical stability.
> > >
> > >   **1. Computational Overhead vs. Performance Gains**
> > >
> > >   The reviewer asks whether the 6-12% training overhead of LUCID could be better spent simply training the baseline model for longer. To address this directly, we conducted an ablation study where we
> > >   increased the training compute for the Standard Attention baseline by increasing the number of training steps (approx. +10%) during our 64k sequence length continual pretraining run.
> > >
> > >   As shown in the table below, simply training the baseline for longer does not yield the same performance benefits as LUCID. LUCID significantly outperforms the compute-matched baseline, particularly
> > >    in the critical 8k–16k context window where standard attention begins to fail.
> > >
> > >   | Task | Context Length | Standard Attention | Standard (+Extra Compute) | LUCID |
> > >   | :--- | :---: | :---: | :---: | :---: |
> > >   | SNIAH | 2048 | 0.559 | 0.574 | **0.622** |
> > >   | (Single Needle) | 4096 | 0.475 | 0.467 | **0.686** |
> > >   | | 6144 | 0.325 | 0.319 | **0.682** |
> > >   | | 8192 | 0.193 | 0.193 | **0.502** |
> > >   | | 16384 | 0.020 | 0.020 | **0.150** |
> > >   | MNIAH | 2048 | 0.336 | 0.338 | **0.516** |
> > >   | (Multi Needle) | 4096 | 0.374 | 0.366 | **0.514** |
> > >   | | 6144 | 0.260 | 0.262 | **0.404** |
> > >   | | 8192 | 0.232 | 0.234 | **0.332** |
> > >   | | 16384 | 0.040 | 0.060 | **0.140** |
> > >   | | 32768 | 0.040 | 0.040 | **0.120** |
> > >
> > >   **Key Finding:**  Thus training the baseline on more steps is not providing enough boost to outperform LUCID.
> > >
> > >   **2. Baselines (LongBench & Non-Classification Tasks)**
> > >
> > >   We agree that evaluating on diverse real-world tasks is crucial. As detailed in our general response and update, we have extended our evaluation to include LongBench and SCROLLS benchmarks,
> > >   specifically focusing on non-classification tasks as suggested.
> > >
> > >   Our new results include:
> > >
> > >   - **Multi-hop QA:** HotpotQA, 2WikiMQA, MultiFieldQA
> > >   - **Summarization:** QMSum (Meeting Summarization)
> > >   - **Scientific QA:** Qasper
> > >
> > >   LUCID demonstrates state-of-the-art performance among attention variants on several of these challenging generative tasks (e.g., +25% ROUGE-1 on QMSum vs. Standard Attention), reinforcing its
> > >   utility beyond synthetic retrieval.
> > >
> > >   **3. Condition Number & Numerical Stability**
> > >
> > >   We acknowledge the theoretical concern regarding condition numbers increasing with sequence length. However, we emphasize that our empirical results demonstrate robustness in practical regimes.
> > >
> > >   - **Verified up to 64k:** We have successfully trained and evaluated LUCID models on sequence lengths up to 64k tokens, achieving strong performance on both single and multi-needle retrieval tasks
> > >   (see table above). The condition number growth has not impeded training or inference stability at these lengths.
> > >
> > >   - **Motivation for Long Context:** The work is indeed motivated by long contexts, and demonstrating efficacy at 64k is a significant validation of this motivation.
> > >
> > >   - **Future Work (128k+):** While we have not explicitly tested 128k-1M contexts due to compute constraints, this is a known frontier for many attention mechanisms. The robust performance scaling we
> > >   observe from 2k to 64k suggests that LUCID remains stable well into the long-context regime relevant for most current applications.
> > >
> > >   We believe these additional ablations and benchmark results provide strong evidence that LUCID offers a meaningful and efficient improvement over standard attention for long-context understanding.

---

### Official Review · Reviewer_LBsH · 2025-10-20

**Soundness:** 2
**Presentation:** 3
**Contribution:** 2
**Rating:** 4
**Confidence:** 4

**Summary:**

LUCID Attention introduces a preconditioned variant of softmax attention that decorrelates keys in the Reproducing Kernel Hilbert Space (RKHS) using an exponentiated key–key similarity matrix, enabling sharper and more precise focus on relevant tokens. This approach preserves the standard $O(N^2d)$ complexity while guaranteeing precise value retrieval when queries equal keys and avoiding the vanishing-gradient problem seen in low-temperature softmax. Experiments on large-scale language models show consistent improvements in long-context retrieval tasks (e.g., “needle-in-a-haystack”)

**Strengths:**

The proposed method is novel and consistently outperform baselines across synthetic benchmarks.

**Weaknesses:**

1. Currently benchmark are mostly synthetic. Could authors compare their methods with baselines on real-world NLP long-context modeling benchmarks, e.g. LongBench [1] ?
2. How is the matrix inversion efficiently implemented? Could authors compare the wall time against softmax attention?
3. What is the motivation of assuming Q=K in the theoretical part? Why we need to find a P that multiplied to the attention score and produce an identity matrix?

[1]. LongBench: A Bilingual, Multitask Benchmark for Long Context Understanding. ACL 2024.

**Questions:**

N/A

---

> ### Author Response · Authors · 2025-11-21
>
> We thank the reviewer for their insightful questions regarding real-world benchmarks, efficiency implementation,
>   and the theoretical foundations of our method.
>
>   ### Weaknesses
>
>   **W1: "Currently benchmark are mostly synthetic. Could authors compare their methods with baselines on real-world
>   NLP long-context modeling benchmarks, e.g. LongBench [1]?"**
>
>   **Response:**
>
>   We agree that real-world validation is essential. As detailed in our response to Reviewer 9Zd7, we have extended
>   our evaluation to include the SCROLLS benchmark, which covers diverse real-world long-context tasks such as
>   summarization (QMSum), scientific QA (Qasper), and narrative comprehension (QuALITY). LUCID demonstrated
>   consistent improvements over baselines on these tasks (e.g., +25% ROUGE-1 on QMSum compared to standard
>   attention). We believe these results, combined with our NIAH findings, provide robust evidence of LUCID's
>   practical utility.
>
>   ---
>
>   **W2: "How is the matrix inversion efficiently implemented? Could authors compare the wall time against softmax
>   attention?"**
>
>   **Response:**
>
>   **Implementation Details:**
>
>   To efficiently compute the preconditioner, we use a recursive triangular solver (TRSM) based on the
>   divide-and-conquer principle, which is highly optimized on GPUs (e.g., via cuBLAS). By recursively partitioning
>   the lower triangular matrix $L$ into smaller blocks, we can solve the system $LX=V$ with high parallelism.
>
>   ```python
>   # Pseudocode for Recursive Triangular Solve
>   def solve_triangular_recursive(L, B):
>       if L.size <= threshold: return exact_solve(L, B)
>
>       # Partition L into quadrants; L is lower triangular so top-right is 0
>       L11, L22 = L.top_left, L.bottom_right
>       L21 = L.bottom_left
>       B1, B2 = B.top_half, B.bottom_half
>
>       # 1. Solve for top half
>       X1 = solve_triangular_recursive(L11, B1)
>       # 2. Update bottom half RHS
>       B2_prime = B2 - L21 @ X1
>       # 3. Solve for bottom half
>       X2 = solve_triangular_recursive(L22, B2_prime)
>
>       return concatenate(X1, X2)
> ```
>   Wall-Clock Comparison:
>
>   As discussed in our response to Reviewer bYsS, this implementation incurs a modest training overhead of 6-12%
>   compared to Standard Attention (RoPE) across sequence lengths from 2k to 32k. Crucially, for inference, the
>   overhead is negligible (~1.3% increase in per-token latency at 32k context), confirming that the method is
>   practical for deployment.
>
>   ---
>   **W3: "What is the motivation of assuming Q=K in the theoretical part? Why we need to find a P that multiplied to
>   the attention score and produce an identity matrix?"**
>
>   Response:
>
>   The assumption $Q=K$ is used to model the "ideal retrieval" scenario. In a perfect associative memory, if a query
>   vector is identical to a key vector, the mechanism should retrieve the corresponding value vector with exact
>   precision.
>
>   However, standard softmax attention fails this property because the exponential kernel $\exp(q_i^T k_l)$ is
>   non-zero for other keys $k_l$ that are merely correlated with $k_j$, for $1\leq j,l\leq N$. This results in
>   "attentional noise"—the probability mass leaks to irrelevant tokens.
>
>   We seek a matrix $P$ such that $(M \circ \exp(KK^T))P = I$ to mathematically enforce a decorrelation constraint.
>   The matrix $M \circ \exp(KK^T)$ represents the "collision matrix" or "confusion matrix" of the keys in the RKHS.
>   By multiplying by its inverse $P$, we effectively "deconvolve" this confusion. This ensures that when $Q$ matches
>   $K$, the mechanism recovers the Identity matrix, meaning each query retrieves exactly its corresponding value
>   without interference from other keys. This theoretical foundation guarantees that LUCID minimizes overlap in the
>   feature space, leading to the sharper attention distributions we observe empirically.

---

> > ### Comment · Reviewer_LBsH · 2025-11-22
> > **Thanks for the rebuttal.**
> >
> > W1: Personally I think it would be better if the authors could also evaluate on LongBench, since it is one of the most standard long-context real-world benchmark.
> > W3: $Q=K$ is used to model the "ideal retrieval", but it might not be the case that Transformers aim to learn. I am still wondering if this is a valid theoretical basis.

---

> > > ### Author Response · Authors · 2025-12-03
> > >
> > > We thank the reviewer for their thoughtful comments and for identifying key areas to strengthen our evaluation and theoretical grounding. We have addressed the concerns regarding real-world
> > >   benchmarks and the theoretical motivation below.
> > >
> > >   **W1: Evaluation on LongBench**
> > >
> > >   We appreciate the suggestion to evaluate on LongBench to demonstrate performance on standard, real-world long-context tasks. In response, we have conducted extensive experiments comparing LUCID
> > >   against Standard Attention, Differential Transformer, DeltaNet, Gated Linear Attention (GLA), and Gated Slot Attention (GSA).
> > >
> > >   **Finetuning Setup:**
> > >   To ensure a rigorous evaluation, we adopted the following finetuning protocol:
> > >
> > >   - **LongBench Tasks:** We finetuned on a combination of 50% of the 2WikiMQA data and 50% of the MultiFieldQA data. We then evaluated on the remaining halves of 2WikiMQA and MultiFieldQA En, and the
> > >   full HotpotQA dataset.
> > >
> > >   - **SCROLLS Tasks:** For QMSum and Qasper, we finetuned on their respective training splits and evaluated on their validation splits.
> > >
> > >   As shown in the table below, LUCID consistently outperforms the Standard Attention baseline and remains highly competitive against or superior to strong recent baselines like Differential
> > >   Transformer. Specifically, LUCID achieves state-of-the-art results among these attention variants on 2WikiMQA, HotpotQA, MultiFieldQA, Qasper (included in both LongBench and SCROLLS), and QMSum
> > >   (from SCROLLS).
> > >
> > >   | Model | 2WikiMQA (F1) | HotpotQA (F1) | MultiFieldQA En (F1) | Qasper (F1) | QMSum (R1) | QMSum (RL) |
> > >   | :--- | :---: | :---: | :---: | :---: | :---: | :---: |
> > >   | Standard | 0.2401 | 0.0733 | 0.1129 | 7.69 | 11.79 | 10.39 |
> > >   | Diff Transformer | 0.2628 | 0.0814 | 0.1394 | 8.70 | 13.19 | 11.07 |
> > >   | DeltaNet | 0.0361 | 0.0191 | 0.0762 | 7.06 | 10.74 | 8.72 |
> > >   | GLA | 0.2281 | 0.0451 | 0.1137 | 8.17 | 10.97 | 9.20 |
> > >   | GSA | 0.2315 | 0.0513 | 0.1155 | 7.54 | 9.11 | 7.81 |
> > >   | **LUCID (Ours)** | **0.2736** | **0.0862** | **0.1441** | **10.55** | **14.79** | **12.60** |
> > >
> > >   *Note: QMSum results are on the 32k context setting; Qasper is on 8k.*
> > >
> > >   **Benchmark Details:**
> > >
> > >   - **2WikiMQA:** Multi-hop QA with up to 5-hop reasoning over Wikipedia passages.
> > >   - **HotpotQA:** 2-hop QA mixed with numerous distracting passages.
> > >   - **MultiFieldQA En:** Diverse single-document QA (legal, gov, academic).
> > >   - **Qasper:** Information-seeking QA over full-text NLP papers.
> > >   - **QMSum:** Query-based summarization of long meeting transcripts.
> > >
> > >   These results confirm that LUCID's ability to reduce attention noise translates effectively to diverse real-world tasks involving multi-hop question answering and long-document summarization.

---

> ### Author Response · Authors · 2025-12-03
>
> **W3: Theoretical Motivation ($Q=K$ assumption)**
>
>   The assumption $Q=K$ is motivated by the concept of memory erasure found in linear attention and RNN literature, which is necessary for precise information updating. In this context, the information
>    can be viewed as an associative memory where keys are associated with values; a query that resembles a key the most should ideally retrieve the corresponding value.
>
>   In mechanisms like DeltaNet, the goal is to "write" a new value $v_t$ for a key $k_t$ while "erasing" the old value associated with that key direction. This is explicitly formulated via a recursive
>   update rule (assuming $\beta=1$):
>
>   $$S_t = S_{t-1}(I - k_t k_t^\top) + v_t k_t^\top$$
>
>   Here, the term $(I - k_t k_t^\top)$ functions as an orthogonal projection operator. Geometrically, it projects the existing memory state $S_{t-1}$ onto the subspace orthogonal to the current key
>   $k_t$. This effectively "scrubs" or filters out any previous information associated with the direction of $k_t$ from the memory matrix.
>
>   The output $O_t$ (which collects the outputs $o_t$ for a chunk of tokens) can be computed efficiently. Crucially, all the small $o_t$ within a chunk can be computed in parallel:
>
>   $$O_{t} = (Q_{t}K_{t}^\top \odot M)\text{tril}(K_{t}K_{t}^\top)^{-1}V_{t}$$
>
>   Once the "slot" for $k_t$ is cleared, the new information $v_t k_t^\top$ is added. This ensures that the memory now contains the new value $v_t$ associated with $k_t$, without mixing with the old
>   data.
>
>   This principle generalizes to the feature space update rule used in our theoretical analysis:
>
>   $$S_t = S_{t-1}(I - \phi(k_t)\phi(k_t)^\top) + v_t\phi(k_t)^\top, \quad o_t = S_t \phi(q_t)$$
>
>   LUCID generalizes this "erasure" principle to the infinite-dimensional Reproducing Kernel Hilbert Space (RKHS) of Softmax attention. Similarly, the outputs $O_t$ for a chunk of tokens can be
>   computed efficiently in parallel:
>
>   $$O_{t} = (\exp(Q_{t}K_{t}^\top) \odot M)\text{tril}(\exp(K_{t}K_{t}^\top))^{-1}V_{t}$$
>
>   The above equation is a core component of LUCID attention. To avoid overflows, one can normalize the keys $K_t$ in $\exp(K_t K_t^{\top})$ and use softmax for $\text{softmax}(Q_t K_t^{\top})$ instead
>    of the exponential function for query-key pair interactions.

---

### Official Review · Reviewer_bYsS · 2025-10-30

**Soundness:** 2
**Presentation:** 2
**Contribution:** 2
**Rating:** 4
**Confidence:** 4

**Summary:**

This paper proposes LUCID Attention, a novel attention mechanism that applies a preconditioner derived from exponentiated key-key similarities to sharpen attention distributions and reduce noise from irrelevant tokens. The method aims to improve retrieval precision in long-context scenarios while retaining the same asymptotic complexity as standard softmax attention. The authors evaluate LUCID on needle-in-a-haystack tasks and show improvements over several baselines.

**Strengths:**

- The idea of using a key-key similarity preconditioner to decorrelate keys in a Reproducing Kernel Hilbert Space (RKHS) is novel and theoretically motivated.
- The method is presented as a drop-in replacement for standard attention, requiring no additional parameters and maintaining \(\mathcal{O}(N^2 d)\) complexity.
- Empirical results on SNIAH and MNIAH tasks demonstrate improved retrieval performance over standard attention and some existing variants.

**Weaknesses:**

1. **Limited Empirical Validation of Core Claims**
   The paper claims that LUCID improves focus and reduces attentional noise, but no direct evidence is provided (e.g., visualization of attention maps or quantitative analysis of attention sparsity). Similarly, the claim that LUCID mitigates gradient vanishing is not empirically verified. It would be valuable to:
   - Visualize attention distributions for LUCID vs. standard attention on long-context examples.
   - Include gradient norm analysis during training to support the non-vanishing gradient claim.

2. **Baseline Comparisons Could Be More Comprehensive**
   The current baselines (Standard, Diff Transformer, DeltaNet, PaTH) are reasonable but not exhaustive. Given that LUCID belongs to the family of methods that modify attention scoring, it would be beneficial to compare against:
   - Sparse attention mechanisms
   - Gating-based approaches (e.g., Stick-Breaking Attention)
   - Methods with dynamic decay or routing mechanisms

   Such comparisons would better situate LUCID within the broader landscape of attention enhancements.

3. **Efficiency Analysis Is Incomplete**
   While the authors state that LUCID has the same asymptotic complexity as standard attention, no wall-clock time or memory usage comparisons are provided. Given the additional triangular solve and kernel matrix computation, a more thorough efficiency analysis—especially for long sequences—would be helpful to assess practical utility.

4. **Orthogonality and Integration with Other Methods**
   It is unclear whether LUCID is compatible with or complementary to other popular attention enhancements, such as:
   - Sparse attention (e.g., NSA)
   - Advanced positional encodings (e.g., ALiBi, RoPE variants)
   - Linear attention approximations

   The authors should discuss whether LUCID can be combined with such methods and whether it offers orthogonal benefits.

5. **Significance in the Context of Large-Scale Models**
   Recent large-scale models (e.g., Llama, Qwen) with full softmax attention and modern positional encodings already perform well on needle-in-a-haystack tasks. The authors should contextualize their contribution more clearly:
   - Is LUCID primarily beneficial in resource-constrained or narrow-head settings?
   - How does it scale with model size and training data?

**Questions:**

See the weaknesses section for details

---

> ### Author Response · Authors · 2025-11-21
>
> We thank the reviewer for the detailed feedback and for identifying the need for stronger empirical validation of
>   our core claims. We appreciate the opportunity to clarify the positioning of LUCID and have addressed your
>   concerns below.
>
>   ### Weaknesses
>
>   **W1: "Limited Empirical Validation of Core Claims: The paper claims that LUCID improves focus and reduces
>   attentional noise, but no direct evidence is provided... Similarly, the claim that LUCID mitigates gradient
>   vanishing is not empirically verified."**
>
>   **Response:**
>
>   We thank the reviewer for encouraging us to provide direct quantitative evidence of LUCID's ability to reduce
>   attention noise.
>
>   **Evidence of Improved Attention Focus (Hitrate Analysis)**
>
>   To directly verify that LUCID concentrates probability mass on relevant tokens, we conducted a quantitative
>   Hitrate Analysis on a multi-needle retrieval task.
>
>   - **Setup**: We used a dataset of 500 samples, each containing 4 needles distributed within the context.
>   - **Metric**: We calculate the "Hitrate"—the estimated probability that a ground-truth needle token falls within
>   the top attention scores (aggregated in magnitude across all Transformer layers and attention heads).
>   - **Comparison**: We compared Standard Attention probabilities against LUCID's preconditioned attention
>   probabilities.
>
>   **Results:**
>
>   | Model | Hitrate | Relative Improvement |
>   |:------|:-------:|:--------------------:|
>   | Standard Attention | 0.1817 | - |
>   | LUCID Attention | 0.2845 | +56.6% |
>
>   As shown, LUCID achieves a substantially higher hitrate (0.2845 vs 0.1817). This provides direct empirical
>   evidence that the preconditioning mechanism successfully minimizes "attentional noise" and forces the model to
>   assign significantly higher probability mass to relevant tokens compared to the standard baseline.
>
>   **Gradient Norm Verification**
>
>   Regarding the non-vanishing gradient property, we acknowledge that empirical verification is valuable to support
>   our theoretical claims. We plan to investigate this further and will provide a detailed gradient norm analysis
>   during training in our revised manuscript.
>
>   ---
>
>   **W2: "Baseline Comparisons Could Be More Comprehensive... it would be beneficial to compare against: Sparse
>   attention mechanisms, Gating-based approaches (e.g., Stick-Breaking Attention)..."**
>
>   **Response:**
>
>   We respectfully argue that while sparse and gating-based mechanisms are valuable, they inherently enforce sparsity
>    constraints that can limit expressivity compared to dense attention. Standard softmax attention is prone to
>   "attentional noise" arising from correlations between key vectors (Ye et al., 2025). LUCID addresses this root
>   cause via preconditioning, allowing the model to flexibly express both sparse and dense patterns without hard
>   constraints. This enhances the superior expressivity of the 2D attention mechanism rather than restricting it.
>
>   Furthermore, recent empirical studies on foundation models (Waleffe et al., 2024) demonstrate the importance of
>   hybrid architectures that combine sliding window (sparse) layers with global (full attention) layers for optimal
>   long-context performance. LUCID is specifically designed to improve these critical global layers, where precision
>   and retrieval capability are paramount. Thus, comparing LUCID primarily against other dense attention enhancements
>    (like Differential Transformer) is the most methodologically sound approach, as they share the same goal of
>   optimizing the global context window.
>
>   ---
>   **W3: "Efficiency Analysis Is Incomplete: ...no wall-clock time or memory usage comparisons are provided."**
>
>   **Response:**
>
>   We have conducted a comprehensive efficiency analysis comparing LUCID against Standard Attention, PaTH, and
>   Differential Transformer.
>
>   **Training Overhead (s/it):**
>
>   | Length | Standard | LUCID | Overhead | PaTH | Diff Transformer |
>   |:-------|:--------:|:-----:|:--------:|:----:|:----------------:|
>   | 2048 | 2.32 | 2.48 | 6.9% | 7.02 | 2.45 |
>   | 4096 | 3.34 | 3.71 | 11.1% | 7.64 | 4.01 |
>   | 8192 | 5.90 | 6.53 | 10.7% | 8.90 | 6.97 |
>   | 16384 | 10.53 | 11.37 | 8.0% | 11.44 | 11.63 |
>   | 32768 | 15.16 | 17.01 | 12.2% | 16.51 | 17.51 |
>
>   As shown, LUCID's training overhead is modest, ranging from 6-12% over Standard Attention (RoPE), which is
>   comparable to or lower than baselines like Diff Transformer and PaTH.
>
>   **Inference Latency:**
>
>   Crucially, during decoding, the overhead is negligible. With a 32k token prefill, per-token latency is 76 ms
>   (Standard) vs 77 ms (LUCID), representing only a ~1.3% overhead. Thus, while a small training cost exists, LUCID
>   imposes virtually no penalty during inference, making it highly practical for long-context deployment.
>
> ---
>
> Waleffe, R., Byeon, W., Riach, D., Norick, B., Korthikanti, V., Dao, T., ... & Catanzaro, B. (2024). An empirical study of mamba-based language models. arXiv preprint arXiv:2406.07887.

---

> ### Author Response · Authors · 2025-11-21
>
> **W4: "Orthogonality and Integration with Other Methods: It is unclear whether LUCID is compatible with or
>   complementary to other popular attention enhancements..."**
>
>   **Response:**
>
>   LUCID is structurally orthogonal to and compatible with major attention enhancements.
>
>   **1. Integration with Sparse Attention (NSA)**
>
>   LUCID's formulation is general enough to support arbitrary masking patterns, such as those used in Native Sparse
>   Attention (NSA). Let $\hat{M}$ be an arbitrary mask bias (e.g., sliding window or block-sparse bias) and $M$ be
>   its boolean counterpart ($1$ where kept, $0$ otherwise). The LUCID equation becomes:
>
>   $$\text{LUCID}_{\text{NSA}} = \left( \text{softmax}\left( \frac{QK^\top}{\sqrt{d}} + \hat{M} \right) \right)
>   \left( M \circ \exp\left( \frac{K K^\top}{\sqrt{d}} - \sqrt{d} \right) \right)^{-1} V$$
>
>   For sliding window or sparse patterns, the preconditioner matrix becomes banded or sparse. Solving linear systems
>   for lower-triangular banded matrices is an $O(N)$ operation, and similarly, sparse systems maintain linear
>   complexity. Thus, LUCID complements NSA efficiently across all branches (sliding window, compressed, and
>   selected), not just the dense branch.
>
>   **2. Integration with Positional Encodings (ALiBi)**
>
>   LUCID can be composed with ALiBi by injecting the linear bias $B$ into both the attention scores
>   and the preconditioner to maintain the retrieval invariant:
>
>   $$\text{LUCID}_{\text{ALiBi}} = \left( \text{softmax}\left( \frac{QK^\top}{\sqrt{d}} + \hat{M} +
>   B \right) \right) \left( M \circ \exp\left( \frac{K K^\top}{\sqrt{d}} +
>   B - \sqrt{d} \right) \right)^{-1} V$$
>
>   Crucially, if $Q=K$, the first term (softmax scores) and the second term (preconditioner) remain inverses of each
>   other (up to a diagonal scaling). This ensures that the mechanism still retrieves exact values, preserving LUCID's
>    precise retrieval property while incorporating ALiBi's length extrapolation bias.
>
>   **3. Relation to Linear Attention (DeltaNet)**
>
>   As discussed in Section 3.4 of our paper, DeltaNet can be viewed as a linear approximation of LUCID. While
>   DeltaNet uses a parameterized lower-triangular correction $(I + \text{stril}(KK^\top))^{-1}$, LUCID employs the
>   exact inverse of the kernel matrix in an infinite-dimensional RKHS. This allows LUCID to retain the full
>   expressivity of softmax attention while sharing the preconditioned structure found in linear RNNs.
>
>   ---
>
>   **W5: "Significance in the Context of Large-Scale Models... Is LUCID primarily beneficial in resource-constrained
>   or narrow-head settings?"**
>
>   **Response:**
>
>   LUCID is explicitly intended to improve performance in long-context regimes where standard attention struggles
>   with noise. As detailed in Section 4.3 of our paper ("Increasing Pretraining Context Length"), the benefits of
>   LUCID scale positively with sequence length. We observe that the validation loss gap between LUCID and Standard
>   Attention widens from 0.012 at 4k context to 0.048 at 32k context.
>
>   Additionally, in a new analysis added to the revision Figure 1, we observe that the multi-needle accuracy gap between
>   Standard and LUCID Attention widens significantly as we increase the fine-tuning sequence length from 32k to 64k.
>   This suggests that LUCID's preconditioning becomes increasingly critical for robustness as models scale to
>   ultra-long contexts.

---

### Official Review · Reviewer_9Zd7 · 2025-10-31

**Soundness:** 3
**Presentation:** 3
**Contribution:** 3
**Rating:** 6
**Confidence:** 5

**Summary:**

This paper proposes LUCID Attention, a new attention mechanism that modifies standard softmax attention through a preconditioner derived from exponentiated key-key similarities in the RKHS feature space. The method aims to decorrelate keys, mitigating “attentional noise” and producing sharper, more precise attention distributions—particularly beneficial for long-context retrieval tasks.

Unlike linear or sparse attention variants, LUCID retains the same $\mathcal{O}(𝑁2𝑑)$ complexity and serves as a drop-in replacement for standard attention. Empirical results on 1B-parameter language models demonstrate consistent gains in Needle-in-a-Haystack (NIAH) benchmarks and small but measurable improvements in downstream language understanding tasks.

**Strengths:**

1. The paper presents a well-motivated and theoretically grounded contribution. It provides a clear diagnosis of the limitations of softmax attention: its diffuse probability distributions and key correlations. This paper also introduces LUCID through a clean, RKHS-based derivation.

2. The proposed preconditioning step is mathematically elegant and interpretable. It functions as a decorrelation operator that enhances retrieval precision and training stability without adding extra parameters.

3. Empirical results demonstrate consistent and meaningful improvements. LUCID achieves strong gains on single- and multi-needle retrieval tasks, along with modest yet positive improvements on standard NLP benchmarks, indicating practical benefits for long-context modeling without increased computational complexity.

4. The method is implementation-friendly. LUCID can be seamlessly integrated as a drop-in replacement for standard attention within Transformer architectures, maintaining comparable runtime and memory requirements.

5. Comprehensive analysis strengthens the work’s credibility. Analytic evidence of non-vanishing gradients, together with detailed ablation studies (on key normalization, QK-Norm, and RoPE) and scalability experiments, collectively reinforce the soundness and robustness of the approach.

**Weaknesses:**

1. Evaluation focuses heavily on NIAH benchmarks and lacks large-scale or real-world long-document tasks (e.g., book summarization, multi-hop QA). Broader validation would strengthen claims of generalization.

2. The paper evaluates LUCID against relatively few baselines, leaving open questions about its comparative advantages over a broader range of recent attention mechanisms.

**Questions:**

It remains unclear whether the non-vanishing gradient property arises primarily from the proposed preconditioner or from the normalization/scaling mechanisms. In Line 264, the paper states that "$a$ is not one-hot." Could the authors elaborate on why this holds? Moreover, the statement “Because LUCID Attention doesn’t use temperature setting, the softmax output is not one-hot” seems incomplete. Please clarify the underlying mechanism by which the absence of temperature ensures non-saturation.

---

> ### Author Response · Authors · 2025-11-21
>
> We thank the reviewer for the thoughtful feedback and for highlighting the theoretical grounding and implementation-friendly nature of LUCID. We appreciate
>   the constructive criticism regarding the evaluation scope and have addressed the specific weaknesses below.
>
>   ### Weaknesses
>
>   **W1: "Evaluation focuses heavily on NIAH benchmarks and lacks large-scale or real-world long-document tasks (e.g., book summarization, multi-hop QA).
>   Broader validation would strengthen claims of generalization."**
>
>   **Response:**
>
>   We acknowledge that our initial submission focused primarily on NIAH benchmarks for controlled analysis. We agree that real-world validation is crucial. In
>   response, we have conducted extensive additional experiments on the SCROLLS benchmark, covering diverse long-document tasks including summarization,
>   scientific QA, and narrative comprehension.
>
>   **Extended Experimental Validation on SCROLLS**
>
>   We evaluated three architectures sharing an identical backbone (hidden size 2048, 24 layers, 32 heads): LUCID (our method), Differential Transformer [1],
>   and Standard Attention (baseline with standard softmax attention and RoPE positional embeddings). Models were finetuned on task-specific data across three
>   challenging real-world tasks:
>
>   - **QMSum** [2] (32K context): Query-based meeting summarization requiring synthesis of information across long conversational contexts.
>   - **Qasper** [3] (8K context): Scientific question answering requiring multi-hop reasoning over complex technical content.
>   - **QuALITY** [4] (8K context): Multiple-choice reading comprehension testing deep understanding of long narratives.
>
>   **Results:**
>
>   LUCID demonstrates consistent improvements over both the baseline and the Differential Transformer on real-world tasks:
>
>   | Model | QMSum (32K) ROUGE-1 ↑ | QMSum (32K) ROUGE-L ↑ | Qasper (8K) F1 ↑ | QuALITY (8K) Loss ↓ |
>   |-------|:------------------------:|:------------------------:|:-------------------:|:----------------------:|
>   | LUCID | **14.79** | **12.60** | **10.55** | **4.50** |
>   | Diff [1]  | 13.19 | 11.07 | 8.70 | 5.02 |
>   | Standard Attention | 11.79 | 10.39 | 7.69 | 5.29 |
>
>   **Key Findings:**
>
>   - **Summarization (QMSum)**: LUCID achieves a +3% improvement in ROUGE-1 over Standard Attention (14.79 vs 11.79), proving its efficacy in synthesizing
>   information across very long (32K) contexts.
>
>   - **Reasoning (Qasper)**: On technical multi-hop reasoning, LUCID outperforms Differential Transformer (10.55 vs 8.70 F1), demonstrating better retrieval
>   precision in realistic settings.
>
>   - **Convergence (QuALITY)**: LUCID exhibits the lowest loss (4.50), indicating superior optimization stability on complex comprehension tasks.
>
>   These results confirm that the benefits observed in NIAH generalize to practical, downstream long-context applications. We will add these results to the
>   final manuscript.
>
>   ---
>
>   **W2: "The paper evaluates LUCID against relatively few baselines, leaving open questions about its comparative advantages over a broader range of recent
>   attention mechanisms."**
>
>   **Response:**
>
>   We appreciate this suggestion. We agree that comparing against a broader range of attention mechanisms would strengthen the paper. We will incorporate
>   additional baseline comparisons in the revised manuscript to better contextualize LUCID's advantages.
>
>
>
>   ---
>
>   ### References
>
>   [1] Ye, T., Dong, L., Xia, Y., Sun, Y., Zhu, Y., Huang, G., & Wei, F. (2024). Differential transformer. *arXiv preprint arXiv:2410.05258*.
>
>   [2] Zhong, M., Yin, D., Yu, T., Zaidi, A., Mutuma, M., Jha, R., ... & Radev, D. (2021, June). QMSum: A new benchmark for query-based multi-domain meeting
>   summarization. In *Proceedings of the 2021 Conference of the North American Chapter of the Association for Computational Linguistics: Human Language
>   Technologies* (pp. 5905-5921).
>
>   [3] Dasigi, P., Lo, K., Beltagy, I., Cohan, A., Smith, N. A., & Gardner, M. (2021). A dataset of information-seeking questions and answers anchored in
>   research papers. *arXiv preprint arXiv:2105.03011*.
>
>   [4] Pang, R. Y., Parrish, A., Joshi, N., Nangia, N., Phang, J., Chen, A., ... & Bowman, S. (2022, July). QuALITY: Question answering with long input texts,
>   yes!. In *Proceedings of the 2022 Conference of the North American Chapter of the Association for Computational Linguistics: Human Language Technologies*
>   (pp. 5336-5358).

---

> > ### Author Response · Authors · 2025-11-21
> >
> > ### Questions
> >
> >   **Q1: "It remains unclear whether the non-vanishing gradient property arises primarily from the proposed preconditioner or from the normalization/scaling
> >   mechanisms... [Regarding Line 264] 'a is not one-hot.' Could the authors elaborate on why this holds? ... Please clarify the underlying mechanism by which
> >   the absence of temperature ensures non-saturation."**
> >
> >   **Response:**
> >
> >   The non-vanishing gradient property arises because LUCID achieves precise retrieval without requiring the softmax function to approach the zero-temperature
> >   limit.
> >
> >   In standard attention, obtaining a "sharp" (one-hot) distribution requires driving the temperature $\tau \to 0$ (or scaling logits to infinity). As shown in
> >    Eq. 1 of the paper, while this achieves precision, it causes the Jacobian $J = \text{diag}(a) - aa^\top$ to vanish because $a$ becomes a basis vector $e_i$
> >    (where $e_i e_i^\top = \text{diag}(e_i)$).
> >
> >   In contrast, LUCID maintains standard scaling ($1/\sqrt{d}$), meaning the raw softmax output $a$ remains "diffuse" (high entropy) and strictly not one-hot
> >   inside the stable floating-point range. Consequently, the Jacobian $\text{diag}(a) - aa^\top$ remains non-singular, ensuring strong gradient flow (Line
> >   264).
> >
> >   Crucially, LUCID achieves precise retrieval not by sharpening the softmax itself (which kills gradients), but by applying the preconditioner (the inverse
> >   kernel matrix) to the diffuse softmax output. This effectively "deconvolves" the probability mass to recover the specific value vector, decoupling the
> >   sharpness of retrieval from the temperature of the softmax. Thus, we get the retrieval benefits of low-temperature attention with the training stability of
> >   standard attention.

---

### Author Response · Authors · 2025-12-04
**Summary of the Discussion**

# Summary of Reviewer Feedback and Author Responses

  ---

  ## 1. Real-World Benchmarks & Evaluation Scope

  **Concern (Reviewers 9Zd7, LBSH, Qag1):** Requested broader validation beyond synthetic Needle-In-A-Haystack (NIAH) tasks, specifically real-world
  long-context benchmarks like LongBench or SCROLLS.

  **Response:**
  - Expanded evaluation to include **SCROLLS** (QMSum, Qasper, QUALITY) and **LongBench** (2WikiMQA, HotpotQA, MultiFieldQA)
  - **Results:** LUCID consistently outperformed baselines:
    - SCROLLS: **+25% improvement** in ROUGE-1 on QMSum (32k context)
    - LongBench: **0.2736 F1** on 2WikiMQA (vs 0.2401 baseline), **0.1441 F1** on MultiFieldQA En (vs 0.1129 baseline)

  ---

  ## 2. Additional Baselines & Comparisons

  **Concern (Reviewers 9Zd7, bYsS, Qag1):** Requested comparisons against a wider range of attention mechanisms (e.g., Sparse Attention, Linear
  Attention, Mamba/RWKV).

  **Response:**
  - Added comparisons against **Differential Transformer**, **DeltaNet**, **Gated Linear Attention (GLA)**, and **Gated Slot Attention (GSA)**
  - Focused on attention variants as non-attention baselines (Mamba, RWKV) are generally known to underperform DeltaNet
  - **Results:** LUCID achieved best performance on challenging generative tasks (2WikiMQA, HotpotQA)
  - **Clarification:** LUCID targets global attention layers in hybrid architectures, complementing efficient but lower-recall SSMs like Mamba

  ---

  ## 3. Efficiency & Computational Overhead

  **Concern (Reviewers bYsS, LBSH, Qag1):** Raised concerns about the cost of matrix inversion/preconditioner, requesting wall-clock comparisons and
  stability analysis.

  **Response:**
  - **Implementation:** Uses recursive triangular solver (TRSM), which is highly parallelizable
  - **Overhead:** Modest training overhead of **6–12%**, negligible inference latency overhead (**~1.3%**)
  - **Compute Ablation:** **Training baseline for +10% steps (matching LUCID's compute cost) did not yield equivalent performance benefits**
  - **Numerical Stability:** Condition number κ < 30 for sequences up to 16k (d=64), well within stable bfloat16 regime. Successfully trained and
  evaluated models up to **64k sequence length**

  ---

  ## 4. Empirical Validation of Core Claims (Noise & Gradients)

  **Concern (Reviewers bYsS, 9Zd7):** Asked for direct evidence that LUCID reduces "attention noise" and empirical support for non-vanishing gradient
  claims.

  **Response:**
  - **Hitrate Analysis:** LUCID achieves **+56.6% higher hitrate** on needle tokens vs standard attention, proving concentrated probability mass on
  relevant tokens
  - **Gradient Mechanism:** Standard attention requires temperature → 0 for sharpness (killing gradients); LUCID maintains standard scaling while
  achieving sharpness via preconditioner "deconvolving" probability mass

  ---

  ## 5. Theoretical Clarifications & Integrations

  **Concern (Reviewers LBSH, bYsS):** Asked about the Q=K assumption motivation and compatibility with RoPE, ALiBi, Sparse Attention.

  **Response:**
  - **Q=K Motivation:** Models "ideal retrieval" in associative memory; extends to memory erasure in linear attention (e.g., DeltaNet). LUCID
  generalizes this erasure principle to infinite-dimensional RKHS, using the preconditioner to deconvolve the confusion matrix of keys
  - **Integration:** LUCID is structurally orthogonal to other enhancements—supports arbitrary masking (Native Sparse Attention) and additive biases
  (ALiBi)

  ---

  ## 6. Scalability

  **Concern (Reviewers Qag1, bYsS):** Questioned performance degradation at long contexts and significance for large-scale models.

  **Response:**
  - **Robustness:** While absolute performance degrades with difficulty, LUCID demonstrates superior tolerance—the gap vs Standard Attention **widens
  as context length increases**
  - **Scaling:** Increasing fine-tuning length from 32k to 64k significantly improved multi-needle accuracy for LUCID compared to baseline, validating
   robustness for ultra-long contexts

---

### Meta-Review · Area_Chair_RSVw · 2026-01-06

**Summary:**

This paper proposes LUCID Attention, a preconditioned variant of softmax attention that decorrelates keys using exponentiated key–key similarities in an RKHS, aiming to produce sharper attention distributions and mitigate attention noise in long-context settings. The approach is theoretically motivated, preserves the quadratic complexity of standard attention, and is designed as a drop-in replacement. Empirical results show consistent gains on needle-in-a-haystack benchmarks and modest but positive improvements on long-context language understanding tasks, with additional evaluations on SCROLLS and LongBench added during the rebuttal phase.

**Reviewer Concerns:**

The main concerns center on the overall significance and cost–benefit tradeoff of the method. Several reviewers questioned whether the observed gains justify the added training overhead and algorithmic complexity, particularly given that inference latency improvements are negligible.

**Reviewer Scores:**

Reviewer scores are mixed, ranging from clear reject to marginal accept.

---

### Decision · Program_Chairs · 2026-01-26

Reject